# Are Traffic Announcements Really Effective? A Systematic Review of Evaluations of Crash-Prevention Communication Campaigns

**Mireia Faus** [1], **Francisco Alonso** [1,*], **Cesáreo Fernández** [2] and **Sergio A. Useche** [1]

1   DATS (Development and Advising in Traffic Safety) Research Group, INTRAS Research Institute on Traffic and Road Safety, University of Valencia, 46022 Valencia, Spain; mireia.faus@uv.es (M.F.); sergio.useche@uv.es (S.A.U.)
2   ITACA (Research in Technologies Applied to Audiovisual Communication) Research Group, University Jaume I, 12071 Castellon, Spain; cesar.fernandez@uji.es
*   Correspondence: francisco.alonso@uv.es

**Abstract:** Communication campaigns are commonly used in the traffic and road safety sector to raise public awareness of the importance of avoiding risky road user attitudes and behaviors. Surprisingly few of these communication campaigns evaluate their effectiveness in a formal and comprehensive manner. The core aim of the present systematic review is to identify the type of studies that evaluate the effectiveness of campaigns in this sector, in order to identify and contrast their main findings. This systematic review followed the PRISMA methodology, by means of which the relevant articles based on the search term were identified. A total of 613 indexed articles were filtered, and a final set of 27 articles directly addressing the issue was analyzed. Search strategies were developed and conducted in WOS, Scopus, NCBI, Google Scholar and APA databases. The selected articles point out that, although advertisements alone have a certain positive effect, their effectiveness is substantially increased if they are accompanied by other preventive measures such as legislation or road safety education. In any case, more evaluations of traffic campaigns are needed to identify which techniques are effective and which are not, and which should therefore be replaced by new methods of behavior modification in future communication campaigns.

**Keywords:** communication campaigns; advertisements; road safety; preventive measures

## 1. Introduction

Although road traffic fatality rates have been reduced considerably, the latest recorded data indicate that 1.35 million people die each year worldwide as a result of traffic crashes [1–4]. In addition, virtually all traffic accidents are directly or indirectly human-induced [5,6]. Multiple preventive measures have been employed to avoid risky behaviors and contributed to the reduction of road accidents, such as traffic regulations and sanctions [7,8], road safety education [9–11] and communication campaigns and advertisements [12]. Especially relevant have been advertising campaigns and, in particular, audiovisual campaigns [13].

Media campaigns have long been used as a tool to promote public health [14–17]. Thus, the purpose of traffic advertisements has been to alert the population to the risks of poor driving with the aim of reducing road accidents and mortality [18–20]. Thus, although the objective has always been the same, communication strategies have had a great evolution over the years [21]. The first pieces of advertisement launched in the mid-20th century were informative, with the use of graphics and colloquial language [22]. Some spots made use of the star-system resource by including celebrities to convince more of the message, while others used animated characters [23]. From the 1990s onwards, more emotional ads began to be used, with a greater visual impact, making use of harsh and realistic images of

traffic accidents [24]. Currently, heterogeneous spots are used that mix emotional, shocking and informative elements, always emphasizing the value of the individual as a generator of change and a fundamental element in the reduction of accidents [25].

Themes have also evolved according to the causes of accidents in each time period. Thus, there are risk factors that have always been addressed, such as alcohol consumption, speeding or helmet use [26–30], while others have been appearing or disappearing according to changes in society and users. A clear example is a rise in the use of electronic devices that generate distractions and are a direct cause of accidents, which have therefore been one of the main elements addressed in the most recent traffic campaigns [31,32]. In this way, several levels of action are currently covered and integrated, such as the role of the user, co-implication, environment, education and training, infrastructure and vehicle engineering [33].

*Are Road Safety Awareness Campaigns Effective?*

Since the first traffic campaigns, thousands of advertisements have been launched worldwide [34]. Governmental and private entities in many countries make use of these road awareness campaigns, in which a lot of material, economic and social efforts are invested [35]. It is, therefore, surprising that there are so few evaluations of how effective they are [18]. The fact that there is no information on the consequences of a traffic announcement makes it challenging to determine whether its effectiveness could be improved and, if so, in what ways. Such evaluations would also make it possible to know the degree of effectiveness of a given communication strategy and determine the evolution of campaigns and the possible habituation effect on the population.

In this sense, the present systematic review aims to examine the scientific articles in which a traffic and road safety campaign has been evaluated, whether they are advertisements broadcast in the media (television, radio, written press) or other types of road awareness campaigns. The importance of this work lies in finding common elements (and discrepancies) that can provide generalizable conclusions that will undoubtedly be useful in the design and development of future traffic campaigns in order to significantly improve their effectiveness.

## 2. Materials and Methods

### 2.1. Approach

The literature review was carried out by means of a systematic and transparent mapping. In this way, the research questions were synthesized, studies were searched, and their quality was assessed, regardless of whether the nature of the research was quantitative or qualitative [36].

The stages of the systematic review were carried out using the method proposed by Arksey and O'Malley [37], and are as follows:

1. Identifying the Research Question,
2. Finding Relevant Studies,
3. Selecting the Studies,
4. Charting the Data and Collating,
5. Summarizing and Reporting the Results.

2.1.1. Step 1: Identifying the Research Question

As previously mentioned, the objective of this systematic review is to identify the number and type of studies that evaluate the effectiveness of traffic campaigns or advertisements in order to identify the main findings in this field of research, as well as the main discrepancies (or concordances) that may have occurred.

No comparisons were made at this stage. Table 1 shows a summary of the selected articles and a summary of their characteristics and main results.

2.1.2. Step 2: Finding Relevant Studies

The review found in this project was carried out using the PRISMA guidelines for systematic reviews [38]. The consulted databases were: Web of Science, Google Scholar, American Psychological Association (APA) and Scopus. In addition, a reference search was also carried out on the basis of other research articles whose subject matter is similar to that of this project. Such a bibliography, of which we have had primary research experience, was selected because it is potentially eligible and could not be captured by our search strategies.

**Table 1.** General characteristics of eligible studies.

| Author and Year | Country | Study Aim(s) and Method | Type of Investigation | Results (Main Outcomes) |
|---|---|---|---|---|
| Negi et al. 2020 [39] | Ethiopia | This study assesses the effectiveness of a public awareness campaign targeted at deterring people from driving while intoxicated. Under the slogan "Never drink and drive", the analysis focuses on knowledge, attitudes and behaviors related to drinking and driving. | Observational and Cross-Sectional | Following the program, participants were far more likely to change their knowledge and attitudes concerning drunk driving. Behavior improved as well: participants after the campaign reported a lower rate of drunk driving compared to those who had participated in the campaign at the beginning. |
| Adnan and Gazder, 2019 [40] | Pakistan | Univariate and non-parametric classification and regression tree (CART) approaches are used to evaluate repeated cross-sectional data obtained before ($n$ = 226) and after ($n$ = 277) the helmet use enforcement campaign. | Observational and Longitudinal | The effect of the campaign is temporary. It takes a lot of effort to make it effective, as well as to establish consistent and systematic awareness and enforcement programs. |
| Sicińska and Dąbrowska-Loranc, 2018 [41] | Polonia | This study presents the results of self-reported data from the real traffic survey on vehicle users' protective behavior. Focusing on their performance following seat belt and child restraint campaigns. | Observational and Longitudinal | Regarding the study results, it should be noted that the annual decrease confirms the need for regular surveys on road user behavior and for educating road users through the media. Such activities encourage motorists to use child and adult restraint systems, which are the most basic and effective passive safety devices in a vehicle. |
| Shaikh et al. 2017 [42] | Pakistan | Three observational surveys on the proportion of vehicles giving way to ambulances on the roads were conducted in different areas of Karachi (Pakistan) for this project, taking place at three different points in time: before, during and after the campaign on the proportion of vehicles giving way to ambulances on the roads. | Observational and Longitudinal | Vehicles were more inclined to give way to ambulances during and after the campaign. Media campaigns, especially when conveying a humanitarian message, can play an essential role in causing a change in people's negligent behavior in response to such activities. |

**Table 1.** *Cont.*

| Author and Year | Country | Study Aim(s) and Method | Type of Investigation | Results (Main Outcomes) |
|---|---|---|---|---|
| Hamelin et al. 2017 [43] | Australia | GfK-face EMO's recognition software collected 60 people's unconscious feelings while they saw the commercial, and they were also requested to complete a modified version of the National Survey of Speeding Attitudes and Behaviors. With this test, a driving attitude score was calculated directly after the individual saw the advertisement and again two weeks later. | Experimental and Cross-Sectional | The highly emotional ad improved drivers' driving behavior. Higher impact than the low-emotional ad. |
| Adamos and Nathanail, 2016 [44] | Greece | The findings of an examination of three road safety communication initiatives targeting the themes of drinking and driving, seat belt use and driving weariness are used in this article. When measuring the success of road safety campaigns, this study uses three types of research designs (experimental, quasi-experimental and non-experimental designs), implements a cross-design assessment and conducts a cross-campaign evaluation. | Mixed and Cross-Sectional | The study's findings revealed that the pre-post split-sample design had better predictability than other designs, particularly in data collected from the intervention group after the campaign's execution. The predictability values increased as more constructions were introduced to the independent variables. The construct that has the greatest impact on behavior is the intention, while the other constructions have a lesser impact. Behavioral beliefs, normative beliefs and descriptive norms all play a role in predicting purpose. |
| Widyastuti et al. 2016 [45] | Indonesia | Many places have utilized road safety campaigns to minimize traffic fatalities and injuries, but few have been created using scientific theories or evaluated for their effectiveness in changing attitudes, intentions or behaviors. The value of the crash rate as a metric to be evaluated is the subject of RUNK's investigation in this study. | Observational and Cross-Sectional | The result of this study shows that road type 4/2 UD has the highest accident rate. |
| Ditsuwan et al. 2013 [46] | Thailand | For this study, a generalized cost-effectiveness analysis was carried out, and costs from a health sector perspective were included. Random and selective breath testing and media campaigns were compared in both the current and intervention scenarios with the "do nothing" scenario. | Observational and Cross-Sectional | When compared to doing nothing, media campaigns, random breath testing and selective breath testing all result in cost savings. Together: intensified breath testing and media campaigns have the potential to reduce the burden of alcohol-related road traffic injuries by 24%. |

**Table 1.** *Cont.*

| Author and Year | Country | Study Aim(s) and Method | Type of Investigation | Results (Main Outcomes) |
|---|---|---|---|---|
| Zampetti et al. 2013 [47] | Italy | The 20 municipalities received a publicity campaign through different actions, such as posters, leaflets, media communication with press conferences, articles in local newspapers, radio and TV interviews and a website dedicated to the LHA1. In addition, 12 municipalities received an intensive education campaign. After that, the number and severity of non-fatal road traffic injuries before and after the campaign were compared. | Observational and Cross-Sectional | The results obtained in this project are in line with other European studies and show that there is a general downward trend, but this is presumably not a direct consequence of road safety education. This does not mean that campaigns are ineffective (they are necessary for raising awareness), but it does indicate that they must be supplemented with additional actions in order to be truly effective. |
| Castillo-Manzano et al. 2012 [21] | Spain | The aim of this project is to study and evaluate the effectiveness of sanctioning strategies in terms of the main indicators of road accidents and the duration of the effects. Multivariate models of unobserved components in a state-space framework are utilized in this fashion on monthly series from 1980 to 2008. | Observational and Longitudinal | The findings of this study are crucial because they show that after several years of soft advertising, when the level of severity of the messages is increased, the number of deaths and injuries is lowered. |
| Hutchinson and Wundersitz, 2011 [48] | Australia | The recent literature on the effectiveness (or otherwise) of road safety promotion through media advertising is selectively reviewed. The overall result is inconclusive: large effects have been rare, but small effects cannot be ruled out. | Observational and Cross-Sectional | It is proposed that the assessment should be based on a before-and-after comparison of objectively observable behaviors or variables that are closely related to security, and credible theories are needed to corroborate the link between behavior and security. |
| Richard and De Barros, 2010 [49] | Canada | The impact of anti-speeding messaging on drivers' attitudes and traffic speed on an interurban highway are investigated in this research. A questionnaire was developed and sampled on almost a hundred drivers. | Observational and Cross-Sectional | The results of the study show that the messages produced do not have a large, albeit beneficial, effect on driver attitudes and road traffic speed. |

**Table 1.** *Cont.*

| Author and Year | Country | Study Aim(s) and Method | Type of Investigation | Results (Main Outcomes) |
|---|---|---|---|---|
| Fell et al. 2008 [50] | United States | The National Highway Traffic Safety Administration (NHTSA) has funded a number of projects in Georgia, Louisiana, Pennsylvania, Tennessee, Texas, Indiana and Michigan. | Observational and Cross-Sectional | Fatal accidents in Georgia, Tennessee, Indiana and Michigan were significantly reduced after using an interrupted time series analysis of FARS data comparing the proportion of alcohol drinking and non-drinking drivers in fatal accidents. In turn, significant reductions were made in a second measure, alcohol-related deaths per 100 million miles traveled per vehicle, in Indiana and Michigan. The other three states showed only minor changes. |
| Solomon et al. 2008 [51] | United States | There were three key components to the Labor Day holiday campaign: (1) DWI enforcement, (2) public awareness efforts and (3) evaluation. The 2006 program used approximately $10 million in congressionally funded audiovisual ads. The message sent out in these campaigns was that drunk drivers would be arrested by the authorities. | Observational and Cross-Sectional | National random-sampling telephone surveys conducted before and shortly after the campaign revealed that the media campaign raised awareness of law enforcement repression and a small increase in the perceived likelihood of being arrested for drinking and driving, but did not reflect self-reported changes in driving behavior while intoxicated. |
| Zwicker et al. 2007 [52] | United States | The state implemented an advertising and implementation model for NHTSA's advertising and enforcement program. This program was conducted in specific counties with the goal of reducing driving and alcohol-related deaths. | Observational and Longitudinal | State DMV surveys in the chosen counties showed a significant increase in information from drivers after the campaign that they had heard of poor driving and had gone through a sobriety checkpoint. Consequently, traffic assessments of drivers' blood alcohol concentrations revealed a considerable decrease in the proportion of drivers with a positive BAC at the end of the campaign when compared to the same period the previous year. Each month, one fatality is expected to be avoided. |

**Table 1.** *Cont.*

| Author and Year | Country | Study Aim(s) and Method | Type of Investigation | Results (Main Outcomes) |
|---|---|---|---|---|
| Whittam et al. 2006 [53] | United States | This study shows the evaluation of a 1/2 month multimedia traffic safety campaign targeting young drivers in Northeast Tennessee. Interviews and discussion groups with young people were carried out to determine the impact on crash frequencies among drivers aged 16 to 19 years, baseline data, intervention and accident monitoring were obtained from the statistics maintained for the state. | Observational and Longitudinal | According to a time series analysis of these data, crashes involving 16–19-year-old drivers reduced by 21.6 percent at the period of intervention, whereas a control area in the southeast of Tennessee revealed no significant changes. |
| Tay, 2005 [54] | Australia | This project evaluated the effectiveness of advertising and law enforcement campaigns against speeding and driving under the influence of alcohol. | Observational and Cross-Sectional | Regarding the results, the advertising and enforcement campaigns against speeding did not show an independent effect. However, its interactive impact was significant in reducing serious accidents involving young drivers. |
| Tay, 2005 [55] | Australia | Drunk driving publicity and compliance campaigns in Victoria have been extensively evaluated. In this sense, the results obtained have generated a great debate. When evaluating the same data from previous studies, this document confirms the effectiveness of the campaigns and tested various assumptions and model specifications. | Observational and Cross-Sectional | After the evaluation, the results obtained were solid and showed the effectiveness of the campaigns to reduce serious accidents during hours of high alcohol consumption. |
| Miller et al. 2004 [56] | New Zealand | In late 1996, nearly a third of the country received alcoholic beverage buses and community alcohol testing programs. This article compares three approaches to the required breath test (CBT), which involves testing all stopped drivers. | Observational and Cross-Sectional | Actions such as aggressive CBT plus zero alcohol tolerance for youth, booze buses and a media blitz proved to be dramatically effective. At the time of carrying out these actions at the same time, night accidents with serious and fatal injuries were reduced by half. Sustained effort appears to be critical. With stepwise, increasingly evident and inevitable checkpoints, better outcomes can be attained than with an "ideal" beginning program. |

Table 1. *Cont.*

| Author and Year | Country | Study Aim(s) and Method | Type of Investigation | Results (Main Outcomes) |
|---|---|---|---|---|
| Agent et al. 2002 [57] | United States | The campaign's assessment phase comprised documenting program-related actions (advertising and compliance) as well as assessing the results. The data from accidents that occurred during the campaign were compared to data from the same period in previous years as part of the review. The number of arrests and other law enforcement activities, as well as telephone polls of drivers done before and after the campaign, and a summary of the sorts of advertising, were all reported. | Observational and Cross-Sectional | The campaign resulted in an increase in the number of drivers who were aware of the program, as well as an increase in the number of drivers who had heard particular details about the campaign. The surveys, on the other hand, found no change in self-reported behavior or a perception of an increased chance of arrest for driving after drinking. |
| Tay and Ozanne, 2002 [58] | New Zealand | This study examined the impact of a fear-based advertising campaign aimed at reducing unsafe driving behavior and fatal accident rates. To carry it out, it was argued that this type of campaign, characterized by touching a chord with drivers, could be very effective. However, the most impressionable population is only a part of it. | Observational and Cross-Sectional | This study showed that fatal accident rates, after the broadcast of the campaign, had been reduced in three groups of drivers: women between 15 and 24 years old, women between 25 and 34 years old and men between 35 and 54 years old. |
| Ulleberg, 2001 [59] | Norway | The present analysis aimed to identify young drivers' subtypes ($n = 2524$) and evaluate how they responded to a road safety campaign. | Observational and Cross-Sectional | After the study, it was found that the subtypes differed in how they evaluated and responded to the road safety campaign. The results of the analysis showed that the campaign seemed to attract the most low-risk subtypes. |
| Tay, 2001 [60] | New Zealand | The Land Transportation Safety Authority (LTSA) implemented an enhanced speed and alcohol control campaign, supported by powerful graphic advertisements on television in October 1995. Macpherson and Lewis (1996, 1998). With the aim of reducing the enormous cost of traffic accidents in New Zealand, estimated at NZ $ 3.3 billion in 1994. | Observational and Cross-Sectional | The results showed that the campaign was effective in reducing the number of serious victims. |

**Table 1.** *Cont.*

| Author and Year | Country | Study Aim(s) and Method | Type of Investigation | Results (Main Outcomes) |
|---|---|---|---|---|
| Macpherson and Lewis, 1998 [61] | New Zealand | In this project, data on campaign advertising exposure and other variables believed to be associated with drunk driving behavior are displayed. In this sense, they were subjected to regression analysis to measure the relationship between the incidence of campaign ads and positive evidence breath tests. | Observational and Cross-Sectional | After the study, it should be noted that only a tenuous relationship was found, since it was important for the success of an advertising campaign that it was linked to the application. |
| Murry et al. 1993 [62] | United States | In this study, an analysis of the impact of a paid advertising campaign that aims to reduce the rate of drunk driving in young people was carried out. To do so, it was investigated utilizing (1) before and after test sample surveys collected from both a campaign and a control location, as well as (2) time-series intervention modeling of monthly traffic accident data collected from both sites. | Observational and Longitudinal | These compatible analyses show collaborative evidence that the advertising campaign was successful and reduced the driving behavior and alcoholism of young men and, consequently, traffic accidents. |
| McLean et al. 1991 [63] | Australia | The NHMRC's Highway Accident Investigation Unit conducted a survey of alcohol in drivers' breath in 1989 to monitor the effectiveness of random breath tests (RBTs) by the police. | Observational and Cross-Sectional | The study found a 40 percent reduction in the proportion of drivers above the legal blood alcohol limit. A de-escalation went hand in hand with the increase in the level of publicity for RBT police operations. Additionally, other factors may have played a role in these significant reductions: advertising followed by a rise in the number of RBT applications. |
| King and Reid, 1990 [64] | United States | Building on the general body of research on fear and persuasion, this study was conducted to address the question of whether threats of physical injury of varying intensity and the focus of injury outcome produce differences in increased fear between individuals, and how fear affects cognitive, evaluative and behavioral responses to public service announcements against alcohol and driving. | Experimental and Cross-Sectional | No differences were found across the treatments in support argumentation, attitude toward the PSAs or intention to drink and drive. |

The mapping of the bibliography has covered, in terms of selection dates, the beginning of the databases previously named and the first part of the month of June 2021.

The keywords that were used to search the bibliography were the following: "road traffic campaigns", "communication campaigns", "advertisements", "publicity", "mobility", "evaluation", "effects", "effectiveness" and "road users". The mentioned words were selected after carrying out a study of different articles in the same field. These are the keywords most frequently used in this type of research.

### 2.1.3. Step 3: Selecting the Studies

The articles that were not selected at this stage were those that were not focused on the objective of our research. Therefore, studies evaluating prevention tools and related measures not directly related to advertising or communication campaigns in traffic and road safety were set aside. Therefore, evaluations of weather training or road safety education programs or other similar resources were not included. In addition, publications that emerged through letters, editorials, speeches, conferences/abstracts or case reports were not included.

The authors conducted an individual-level evaluation of a subset of topics and abstracts to subsequently meet and discuss the usefulness of each piece of content, as well as to resolve any discrepancies.

### 2.1.4. Step 4: Charting the Data

After assessment by the authors, the articles that met the inclusion criteria were critically reviewed using the descriptive-analytic method of Arksey and O'Malley [37].

The following information was retrieved and recorded for each selected article: author(s), year of publication, country of study, study design, user group analyzed, sample size, main findings and highlighted results (as shown in Table 1).

### 2.1.5. Step 5: Collating, Summarizing and Reporting the Results

The graphical data were placed in tables and summarized. In addition, they were complemented with descriptive data, analyzed through a thematic organizational strategy (Table 1).

### *2.2. Ethics Statement*

This study was approved by the Ethics Committee of the University Research Institute on Traffic and Road Safety (INTRAS) at the University of Valencia (IRB approval number HE0002020621), certifying that the study design and protocols responded to the general ethical principles applicable for this type of research.

## 3. Results

### *3.1. Search Results*

The keywords used for the literature search identified a total of 613 possible articles. From all of them, discarding of certain research was carried out based on the elimination of duplicate or non-accessible items. In addition, observations were carried out to manually select articles that fit the objective of the present review, leaving 27 eligible articles. Figure 1 shows the data source search and selection process.

### *3.2. Characteristics of Eligible Research Articles*

Although no time limits were placed on the search, the 27 studies that met the inclusion criteria were published between 1990 and 2020. Most of them were published in English ($n = 26$), except for one in Spanish. The studies were conducted in 13 countries, with five continents represented. However, most of the research during the late 1990s and early 2000s concentrated in the United States ($n = 7$), and the oceanic countries, Australia ($n = 5$), and New Zealand ($n = 4$). Meanwhile, most of the remaining countries began their research on the effectiveness of road advertising campaigns after 2010: Pakistan ($n = 2$), Indonesia ($n = 1$), Thailand ($n = 1$), Ethiopia ($n = 1$), Poland ($n = 1$), Greece ($n = 1$), Italy ($n = 1$), Spain ($n = 1$), Norway ($n = 1$) and Canada ($n = 1$).

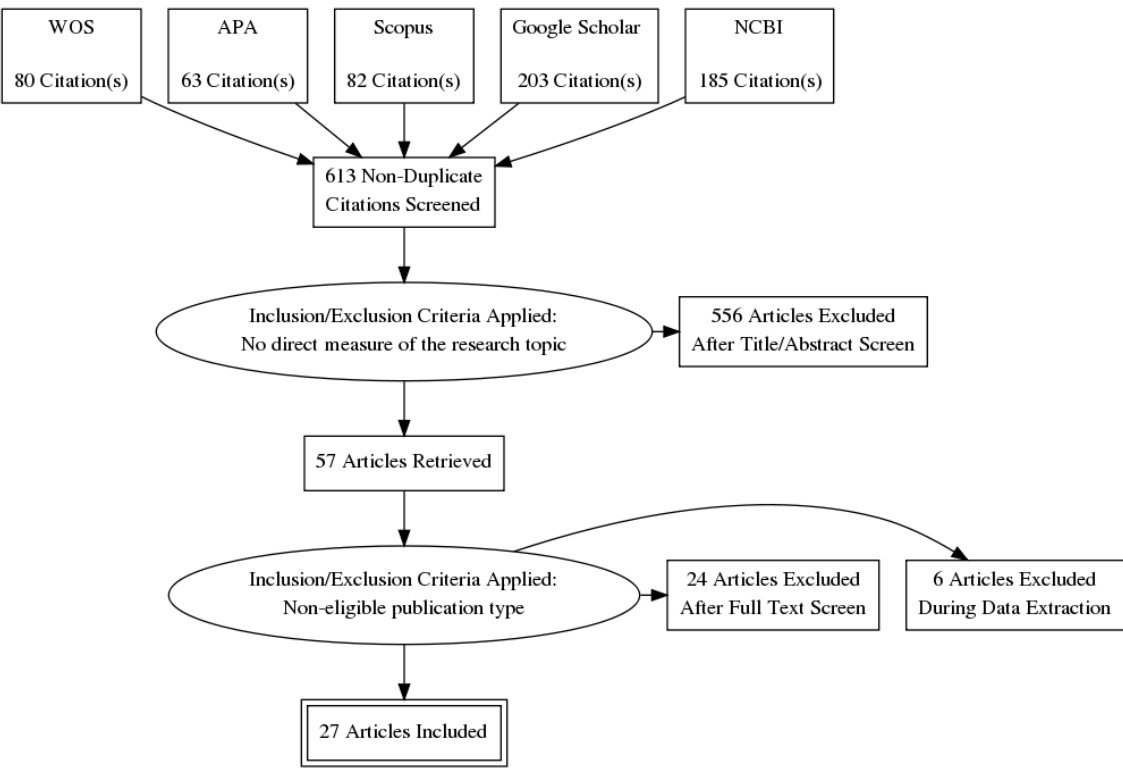

**Figure 1.** PRISMA diagram. Abbreviations: WOS (Web of Science); APA (American Psychological Association).

Most of the evaluations used a non-experimental or observational methodology (*n* = 24). Surveys during and after the broadcasting of the campaigns, focus groups and structured interviews, as well as the frequency of the behaviors targeted by the campaigns (e.g., alcohol and drug consumption, seat belt use, etc.) and accident and traffic fatality data were the most used tools to conclude the degree of effectiveness of the advertisement. Only two of the evaluations made use of the experimental method, where the researcher manipulated the relevant variables to check whether the expected changes occurred (or not). In addition, one of the studies used a mixed methodology (experimental, quasi-experimental and observational) in order to determine the most valid methodological design.

The communication campaigns evaluated also show variability in terms of content. About one-third of the campaigns had among their objectives or as their main objective the reduction of drinking and driving (*n* = 10), concluding that it was effective in reducing this risky behavior, although not to the same extent in all cases. Coincidentally, the most recent campaigns were the ones that obtained the best results (e.g., [39,46]), while those with reduced effects were mostly carried out in the 1990s (e.g., [61,64]). Some advertisements were aimed at the use of passive prevention measures, such as helmets, seat belts and child restraint systems (*n* = 2). In both cases, a positive albeit temporary effect on changing user behavior was observed [40,41]. In campaigns where speeding was the predominant theme (*n* = 3), small effects were observed [49,54], which increased considerably when the ad included high levels of emotion and impact [43].

Other themes included in the evaluated campaigns were the predisposition to ambulance passing [42], and driver fatigue [44], showing positive effects during and after the broadcast. The rest of the campaigns were more generic in that they did not put the focus on a specific risk factor but on several of them (*n* = 10). This block, in general, found a certain trend in the reduction of accident rates, although not only due to the advertising campaign (e.g., [47,60]). In addition, they manifested the clear effect of harshness in advertisements, associating it with greater effectiveness [21].

## 4. Discussion

The present systematic review aims to know the number and type of scientific articles that evaluated traffic and road safety awareness campaigns that have been applied and their main findings. Undoubtedly, the media have a fundamental role in transmitting ideas that increase collective well-being to the population. In this block, we can include the traffic and road safety advertising campaigns [65], with the tools of social marketing being the ones used to achieve awareness [66–68].

The first point to be commented on is the scarcity of articles found. Being a subject with a long history, only 27 articles were found that fit the search criteria. Although this was something we were already aware of, it is surprising that campaigns that take such high economic, personal and time resources from governmental institutions and private entities are not formally and exhaustively evaluated to determine whether they are really effective [18]. It is often assumed that providing information to the user will automatically lead to changes in their behavior, but this does not have to be the case. There are multiple factors involved in this process, such as the way the information is conveyed or the viewer's personality [69].

In this regard, it is especially important to know what to inform about and how to inform. In addition, the communication strategy and even the resources used will probably have to be modified depending on the target audience we are addressing at any given time [70]. Therefore, it is essential to evaluate the effectiveness of campaigns, as it will provide information on which communication strategies are more appropriate, the way to address different types of audience, as well as to know the points on which to place more emphasis and those that are already integrated with the user [71]. The more evaluations that are conducted in this area, the more information will be known to design targeted and effective awareness campaigns.

With this in mind, a question arises: why are road communication and awareness campaigns not evaluated? One possibility is the difficulty in conducting such an evaluation. Crash reduction at a given point in time is often related to a variety of factors [72]. Therefore, it is difficult to evaluate the weight of a campaign when it is developed together with other preventive measures such as road safety education programs or changes in legislation [73]. In this sense, some studies suggest the establishment of several of these measures at the same time has more positive consequences than when these same measures are carried out independently [46]. Another possible reason for not evaluating communication campaigns is that it requires extra financial expenditure. However, being such a recurrent and common tool, that small added effort can provide very relevant information that would allow improving the following campaigns, thus reducing their actual cost [46,74]. In addition, investment in designing more effective campaigns also reduces the economic and personal costs resulting from traffic accidents [75,76]. Therefore, this is a necessary and ultimately recoverable investment.

### 4.1. Effectiveness of Advertising Campaigns

Focusing on the evaluation of traffic and road safety advertising campaigns, the selected articles show considerable variability in the degree of effectiveness of the analyzed campaigns. In some cases, significant changes in relevant behaviors and/or attitudes are observed after exposure to the communication campaign (e.g., [39,42]). However, other research finds no positive effects of traffic ads (e.g., [64]), or very slight ones (e.g., [49]). In any case, it seems clear that the greatest impact is achieved when communication campaigns work in support of other measures such as road safety education programs or increased penalties [46]. This is congruent with the systematic review conducted by Staton et al. (2016) [12], which concludes that low- and middle-income countries use legislation as the preventive tool, but that its effectiveness would increase significantly if they combined such a measure with interventions in the form of road safety education programs and/or communication campaigns.

It should be emphasized that recalling a campaign is not necessarily accompanied by actual behavior change; thus, it is possible that actual behavioral modifications may occur (or not) as a consequence of a recalling campaign, making this not their essential focus [57]. In this regard, various empirical studies highlight that road users engaging the most in risky behaviors, as well as those having a lower road risk perception, are usually the ones less likely to modify their behavioral patterns, especially if they are rather risk-prone [77,78]. On the other hand, the low-risk driver profile is the one that usually most adapts to the norms exposed whether in advertisements, or in other potential information sources [59]. Therefore, it is necessary to propose communication strategies that capture users' attention and manage to convince them of the need not to engage in risky behavior on the road, regardless of their driver profile [71].

In this sense, and given that studies keep several differences in terms of the nature and type of the issues they consider as "key factors to intervene" through a communicative campaign, one must ask about what variables, features and/or approaches should be taken into account to improve the effectiveness of an advertisement. Apparently, a key element is the emotionality of the audiovisual piece [31]. Exposing the viewer to crude, realistic and even violent scenes has shown positive effects because the user can empathize and put himself in the victim's place [79]. However, we must be careful because prolonged exposure to scenes of certain violence can have the opposite effect as they produce habituation. Therefore, the ideal would be to increase the level of harshness after a prolonged period of moderate advertising [21]. Therefore, more aggressive campaigns should be complemented with other types of resources that maintain the emotional and informative element but are not so visually striking [80]. In this sense, different narrative resources can be used, such as reflective, creative and even humorous ads, to make the campaign effective. This makes the viewer consider a change in their driving behaviors and attitudes behind the wheel [81].

Another important factor that is exposed in some of the analyzed articles is the time of effectiveness of a communication campaign. In general, the effect is time-limited [40]. This is consistent with the idea that communication campaigns are usually more effective when they are of short duration and tied to a specific measure [22]. A clear example is holiday periods where sanctions and increased policing are supported by a high frequency of preventive messages in the media.

### 4.2. Methodology for Campaign Evaluation

Great variability was found in the methodology employed in the selected articles, making use of both observational (e.g., [39,48]) and experimental methods (e.g., [43]), as well as cross-sectional (e.g., [46]) and longitudinal ones (e.g., [40]).

In this regard, of particular interest is the analysis by Adamos and Nathanail (2016) [44], in which they make a comparison of the experimental, quasi-experimental and non-experimental (or observational) methods. Their conclusion is clear: the pre–post design is the one that demonstrates better predictability. They add that the more constructs are included as independent variables, predictability values become more reliable, with the "intention" factor being the one best at predicting the change in user behavior.

### 4.3. Practical Implications of the Study

This review is a first evaluation of the strategies used in measuring the effectiveness of traffic communication campaigns. It brings together research from around the world with the purpose of determining the best way to evaluate such campaigns. The scarcity of studies and the variability of results highlight the need to continue this line of research in future traffic campaigns. This is the only way to understand what factors are key for an advertisement to change risk behavior, thus establishing certain guidelines and strategies to improve communication with users through social media.

## 5. Conclusions

The results of this systematic review point to the problematic paucity of literature on the formal evaluation of communication campaigns in the traffic and road safety sector. This means that there is little information on the effectiveness of campaigns in isolation. Evaluation is especially important to detect which commonly used strategies and practices are ineffective. Hopefully, as evaluations become more common in this sector, there will also be developments so that new methods of behavior modification will come into play instead of continuing to use techniques that have not yet proven to be effective [18]. Therefore, the knowledge provided by the existing evaluations collected in this systematic review, as well as those that will be conducted in the coming years, will provide data of interest for the design and development of future traffic and road safety communication campaigns.

**Author Contributions:** For this study, F.A. and M.F. conceived and designed the research, and performed the data collection; S.A.U. and M.F. analyzed the data; C.F. and S.A.U. contributed with reagents/materials/analysis tools; M.F. and S.A.U. wrote and revised the paper. All authors have read and agreed to the published version of the manuscript.

**Funding:** This work was supported by the research grant ACIF/2020/035 (MF) from "Generalitat Valenciana". Funding entities did not contribute to the study design or data collection, analysis, interpretation, or in writing the manuscript.

**Institutional Review Board Statement:** The study was conducted according to the guidelines of the Declaration of Helsinki, and approved by the Institutional Review Board (or Ethics Committee) of the Research Institute on Traffic and Road Safety (IRB number: HE0002020621, from 2 July 2021).

**Informed Consent Statement:** Not applicable.

**Data Availability Statement:** The data will be available upon reasonable request to the corresponding author.

**Acknowledgments:** The authors wish to thank Víctor Palacio and Mayte Duce for the revisions and to Arash Javadinejad (licensed translator) for the professional edition of the final version of the manuscript.

**Conflicts of Interest:** The authors declare no conflict of interest.

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
