# Peer review of "Are Traffic Announcements Really Effective? A Systematic Review of Evaluations of Crash-Prevention Communication Campaigns"

_safety, 2021_

Round 1
Reviewer 1 Report
This article is a good attempt but I think that the authors should add more references to the article. the 70 references are quite short. Please add 15 - 20 more references in your article and correct the English.
Author Response
(Please see the response letter attached in PDF)
Best regards!

Reviewer 2 Report
This work is to identify the type of studies that evaluate the effectiveness of campaigns in this sector, in order to identify and contrast their main findings. The topic is interesting. However, its quality can be improved by addressing the following comments.
- The significance and novelty of this work should be stressed in the introduction.
- I would like to see more details about the methodology of evaluating the effectiveness of the campaigns. Only mentioning the observational and experimental methods, as well as cross-sectional and longitudinal ones is not clear enough.
- If possible, meta-analysis may be conducted to synthesize the evidence across studies to detect effects of the campaigns on the road user attitudes and behaviors.
- Are there any studies on the effective of campaigns on the risk perception of road users? Risk perception is an important factor that influences drivers' behavior and attitude, such as the work of Li. et al.. (2021).
Integration of Theory of Planned Behavior, Sensation Seeking, and Risk Perception to Explain the Risky Driving Behavior of Truck Drivers - The practical implications of this work should be stated in a separate section.
Author Response

(The authors gave the same response as above.)

Reviewer 3 Report
Dear Authors,
The manuscript presented for review in an interesting way presents the assessment of the effectiveness of the undertaken communication campaigns, i.e. preventive measures in the field of road safety. The aim of the presented systematic review was to identify studies that evaluate the effectiveness of communication campaigns. The choice and description of the method is transparent and does not raise any objections, although the small number of studies on the topic presented is surprising. The discussion was carried out in accordance with the research goal, in an interesting and comprehensive way. The conclusions from the presented systematic review undoubtedly have a cognitive value. The overview provides valuable information for the design of road safety preventive measures. The reference list contains 63 correctly selected items, but those available online must be supplemented with the access date.
Best Regards
Author Response

(The authors gave the same response as above.)

Round 2
Reviewer 1 Report
Accept
Reviewer 2 Report
The authors did a good job in addressing my comments. The quality of this manuscript was improved remarkably.
Reviewer 3 Report
Dear Authors,
thank you for the changes.
Best Regards!